# Efficacy, safety and complications of autologous fat grafting to the eyelids and periorbital area: A systematic review and meta-analysis

Fan Yang[1], Zhaohua Ji[2], Liwei Peng[3], Ting Fu[2], Kun Liu[2], Wenjie Dou[4], Jing Li[1], Yuejun Li[1]*, Yong Long[2]*, Weilu Zhang◉[2]*

1 Department of Plastic Surgery and Burns, Tangdu Hospital, The Fourth Military Medical University, Xi'an, China, 2 Department of Epidemiology, Ministry of Education Key Lab of Hazard Assessment and Control in Special Operational Environment, School of Public Health, The Fourth Military Medical University, Xi'an, China, 3 Department of Neurosurgery, Tangdu Hospital, The Fourth Military Medical University, Xi'an, China, 4 Department of Plastic Surgery, Xijing Hospital, The Fourth Military Medical University, Xi'an, China

* zhangweilu@126.com (WZ); longyong71@163.com (YL); liyj@fmmu.edu.cn (YL)

## Abstract

### Background

In recent years, autologous fat grafting (AFG), also known as fat transfer or lipofilling, has been widely performed for periorbital rejuvenation and defect correction, although the evidence regarding its efficacy and safety is still lacking. Besides, with respect to the periorbital region, it is invariably the earliest appearance area of the facial aging phenomenon. Therefore, a systematic review and meta-analysis is needed to evaluate the efficacy and safety of this technique.

### Methods

A literature search was performed in PubMed, Embase, and the Cochrane library databases on November 20, 2020, adhering to the PRISMA guidelines, to identify all relevant articles. Then, a data extraction and standardization process was performed to assess all outcome data. Ultimately, the data were assessed using a random effects regression model with comprehensive meta-analysis software.

### Results

Thirty-nine studies consisting of 3 cohorts and 36 case series with a total of 4046 cases were included. Meta-analysis revealed a relatively high satisfaction rate of 90.9% (95% CI, 86.4%–94.0%). Frequent complications in 4046 patients receiving AFG were edema, chemosis, and contour irregularity, with an overall complication rate of 7.9% (95% CI, 4.8%–12.8%).

**Data Availability Statement:** If the data are all contained within the paper and/or its Supporting information files.

**Funding:** This study was funded by National Natural Science Foundation of China (81773488, 81772096 and 81803289).

**Competing interests:** No authors have competing interests.

## Conclusion

This systematic review and meta-analysis showed that AFG for rejuvenation of eyelids and periorbital area provided a high satisfaction rate and did not result in severe complications. Therefore, AFG might be performed safely for periorbital rejuvenation and reconstruction.

## Introduction

Autologous fat grafting (AFG), also known as fat transfer or lipofilling, is a minimally invasive technique improved by Coleman [1, 2] more than two decades ago. It has been widely used in plastic surgery for various purposes, including restoring contour deformities in patients with sunken upper eyelids [3–7] and promoting skin rejuvenation owing to age-related problem in the periocular region [8]. As for the underlying mechanism, perhaps it is the adipose-derived stem cells (ADSCs) that stimulate angiogenesis and tissue regeneration through secretion of a broad range of cytokines and growth factors [9, 10]. Furthermore, this technique can be used in isolation but it is usually combined with other surgical techniques, for instance, lower eyelid blepharoplasty [11]. However, despite its popularity in periorbital rejuvenation and reconstruction surgeries, it is still questionable whether such a technique that focuses on treating the periorbital problems can be safe, reliable, and effective enough. To date, few studies have conducted randomized controlled trials (RCTs) mostly owing to ethical or practical restrictions. Our understanding of AFG is based on the fragmented knowledge with low quality derived from a case report or case series. A couple of reviews in the past have reached unanimous conclusions; i.e., AFG for periorbital rejuvenation and reconstruction is instructive but unconvincing to some extent [12–15]. Therefore, a thorough synthesis and scientific evaluation based on the published literature on AFG in the form of a meta-analysis should be performed.

## Methods

### Search strategy

The research objectives were to identify, assess, and synthesize the evidence examining the efficacy and safety of AFG in the periocular area. This review was performed in accordance with the PRISMA guidelines [16]. This comprehensive, reproducible, and electronic search was performed via the combination of PubMed, Embase, and Cochrane Library databases. The following keywords were used: [("fat grafting" OR "lipograft" OR "lipoinjection" OR "lipotransfer" OR "fat transfer" OR "fat transplant" OR "lipostructure" OR "lipofilling" OR "fat injection" OR "lipomodeling" OR "fat transplantation") AND ("eyelid" OR "periocular")]. A systematic database search was carried out before November 20, 2020. There were no restrictions with respect to language.

### Inclusion and exclusion criteria

Selected studies met the following criteria: (1) clinical trials of all designs, from the highest level of evidence from randomized trials (if available) to prospective or retrospective observational studies (case series: at least five cases) involving patients receiving AFG for periorbital rejuvenation and reconstruction; (2) the treatment used for periorbital rejuvenation and reconstruction was stated clearly; (3) the study stated the concrete data of postoperative effects; (4) studies with complete follow-up (at least 3 months). Exclusion criteria were as follows: (1) patients with a history of other eyelid surgery or treatment; (2) reviews, letters, commentaries,

reply, discussion, and so on; (3) studies with incomplete or ambiguous or overlapped data; (4) studies not related to the objective of this review.

### Data collection

Two independent reviewers scrutinized the titles, abstracts, and full text of the retrieved articles. If there was any disagreement between the two reviewers, another independent investigator was consulted to reach a consensus. Moreover, a blinded method was used to ensure quality. Data extracted from the eligible articles included the following parts: authors, date of publication, place of study, number of patients, ages of patients, indications, AFG techniques, follow-up time, study design, evidence level, complications, anesthetic evaluation, and satisfaction rates. Then a data extraction sheet was set in Excel (Microsoft, Redmond, Washington, USA). Additionally, each article was assessed for the risk of bias in accordance with the methodological standards listed in the non-comparative case series checklist (for case series) and Ottawa-Newcastle Scale (for cohort studies), respectively (https://www.ncbi.nlm.nih.gov/books/NBK35156/) (S1 and S2 Tables in S1 File).

### Statistical analysis

A meta-analysis of the data from 39 included studies was performed by a comprehensive meta-analysis software, version 2.2.050 (Biostat, Englewood, NJ, USA), and a heterogeneity analysis was conducted for the eligible studies, defining $p<0.05$ as statistically significant. Heterogeneity across studies was calculated by the $I^2$ statistic, with $I^2$ over 50% considered as high heterogeneity. Thus, a random effects model was used to analyze studies with high heterogeneity. Otherwise, the fixed effects model was used. The dichotomous variables were summarized by the Mantel-Haenszel method and compared using relative risks and 95% confidence intervals (CI), which were obtained from a forest plot, and the publication bias was assessed from a funnel plot. To minimize heterogeneity among studies, subgroup meta-analysis was performed among different indications and different result evaluation methods and so on.

## Results

### Literature search

The literature search, performed by using predefined search terms, yielded 839 records. PubMed, Embase, and Cochrane library databases identified 403, 423, and 13 articles, respectively. After all duplicates were removed with the help of the software "Endnote", 604 potential articles were available for screening by reading the titles and abstracts according to the inclusion and exclusion criteria, and 522 articles were eliminated. Then, 82 articles were screened for full text reading. Then based on full-text assessment, we excluded 8 papers that reported studies including less than 5 cases, 16 papers that lacked adequate number of quantitative indicators, and 19 papers that were studies with irrelevant content. Finally, 39 studies were included in this systematic review and were used for quantitative synthesis. The selection criteria and data collection process are shown in Fig 1. Sample sizes of these 39 studies ranged from 5 to 978 and constituted a total sample size of 4046.

### Characteristics of the included studies

Thirty-nine studies were included in this systematic review. Most of the studies included in this systematic review were retrospective, consecutive, nonrandomized interventional case series, except for three retrospective cohort studies [n = 3 (7.69%)] and two prospective studies

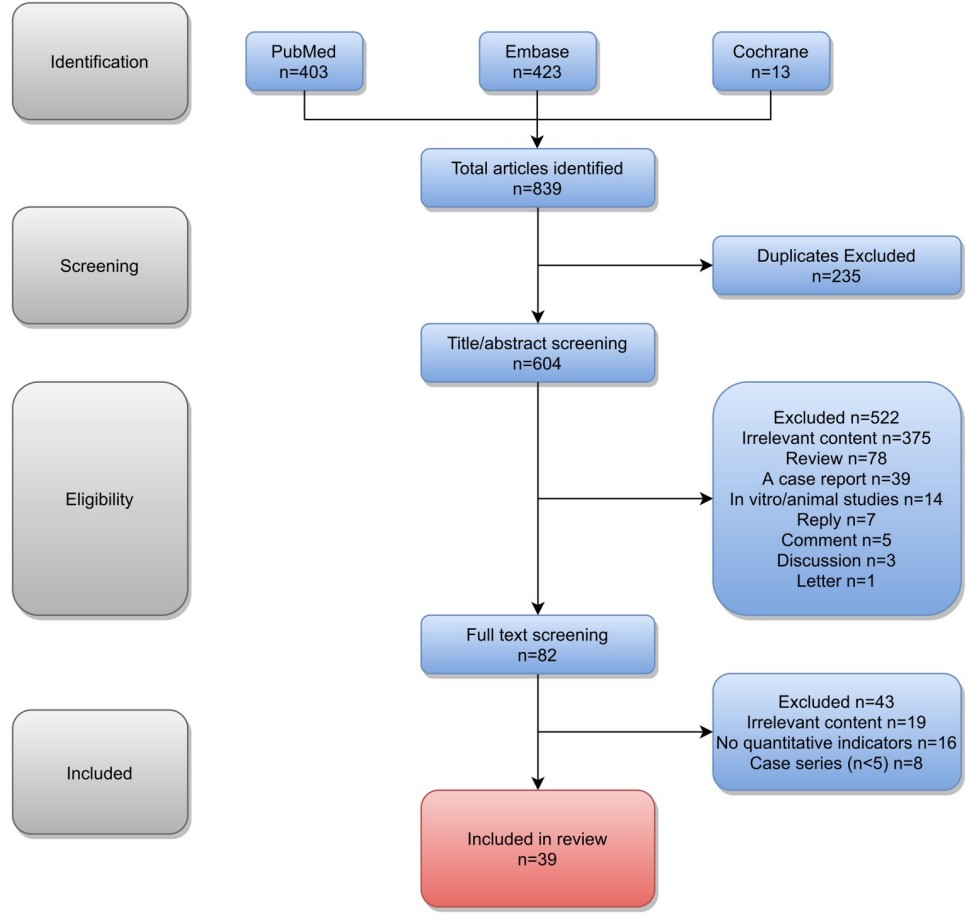

**Fig 1. Flow diagram of the article selection process for review.**

[n = 2 (5.12%)]. Two studies provided relevant control groups to allow for comparison of the results of AFG treatment with control treatment. As most of the studies did not have a control group, no direct comparison between AFG and controls could be made in the meta-analysis. Although most of the included studies were retrospective analyses, the vast majority of studies had included consecutive patients treated with fat grafting, thus decreasing the risk of selection bias to some extent. According to the Oxford Center for Evidence-Based Medicine 2011 guidelines, the levels of evidence were III (3 studies) and IV (36 studies) (Table 1). Furthermore, five studies reported volume-related results to evaluate the volume retention of grafted fat over time, but the data were insufficient to allow for pooling into a meta-analysis. Pelle-Ceravolo et al. [17] reported that 12 of 79 patients had some degree of volume depletion from the 6-month to the 1-year visits, whereas Essuman et al. [18] found that 14 (93.3%) patients had good maintenance of orbital volume at the 3-month follow-up. Also, according to Lee et al. [4], during the 13-month follow-up period, the resorption rate for the dermofat graft was approximately less than 10%–20%. Moreover, three-dimensional imaging was used by Meier et al. [19] to obtain quantitative volume measurements. He concluded that approximately 32% of the injected volume persisted at 16 months. Bernardini et al. [20] reported that volume restoration was regarded by two senior authors as good (63%) and excellent (37%).

**Table 1. General presentation of included articles.**

| Study | Number of patients (Male/Female) | Study design | Level of evidence |
|---|---|---|---|
| Zhou, X.2020 | 38(3/35) | retrospective study (case series) | 4 |
| Pelle-Ceravolo, M.2020 | 200(8/192) | retrospective study (case series) | 4 |
| Lee, W.2020 | 50(7/43) | retrospective study (case series) | 4 |
| Jiang, L.2020 | 50(49/1) | retrospective study (case series) | 4 |
| Biglioli, F.2020 | 75(28/47) | retrospective study (case series) | 4 |
| Larsson, J. C.2019 | 33(6/27) | retrospective study (case series) | 4 |
| Kim, H. S.2019 | 229(65/164) | retrospective study (case series) | 4 |
| Huang, S. H.2019 | 205(22/183) | retrospective study (case series) | 4 |
| Al-Byti, A. M.2019 | 22(0/22) | retrospective study (case series) | 4 |
| Stein, R.2018 | 113(NR) | retrospective study (case series) | 4 |
| Rohrich, R. J.2018 | 131(121/10) VS100(92/8) | retrospective cohort study | 3 |
| Litwin, A. S.2018 | 29(7/22) | retrospective study (case series) | 4 |
| Kim, J.2018 | 978(NR) | retrospective study (case series) | 4 |
| Chen, H.2018 | 9(6/3) | retrospective study (case series) | 4 |
| Ramil, M. E.2017 | 32(0/32) | retrospective study (case series) | 4 |
| Miranda, S. G.2017 | 32(10/22) | retrospective cohort study | 4 |
| Ma, Z.2017 | 32(7/25) | retrospective study (case series) | 4 |
| Lee, W.2017 | 60(9/51) | retrospective study (case series) | 4 |
| Gennai, A.2017 | 65(7/58) | retrospective study (case series) | 4 |
| Chiu, C. Y.2017 | 51VS50 | retrospective cohort study | 3 |
| Skippen, B.2016 | 10(1/9) | retrospective study (case series) | 4 |
| Lin, T. M.2016 | 34(30/4) | retrospective study (case series) | 4 |
| Karataş, M. Ç2015 | 17(11/6) | retrospective study (case series) | 4 |
| Bernardini, F. P.2015 | 98(6/92) | retrospective study (case series) | 4 |
| Lin, T. M.2014 | 168(2/166) | retrospective study (case series) | 4 |
| Le, T. P.2014 | 17(5/12) | retrospective study (case series) | 4 |
| Essuman, V. A.2014 | 15(7/8) | prospective study (case series) | 4 |
| Youn, S.2013 | 82(23/59) | retrospective study (case series) | 4 |
| Tonnard, P. L.2013 | 500(60/440) | retrospective study (case series) | 4 |
| Einan-Lifshitz, A.2013 | 57(10/47) | retrospective study (case series) | 4 |
| Bernardini,F. P.2013 | 400(63/337) | retrospective study (case series) | 4 |
| Park, S.2011 | 50(2/48) | retrospective study (case series) | 4 |
| Chang, H. S.2011 | 8(3/5) | retrospective study (case series) | 4 |
| Roh, M. R.2009 | 10(2/8) | retrospective study (case series) | 4 |
| Meier, J. D.2009 | 33(1/32) | prospective study (case series) | 4 |
| de la Cruz, L.2009 | 34(1/33) | retrospective study (case series) | 4 |
| Korn, B. S.2008 | 11(NR) | retrospective study (case series) | 4 |
| Lee, Y.2001 | 13(NR) | retrospective study (case series) | 4 |
| Malet, T.2000 | 5(NR) | retrospective study (case series) | 4 |

NR, not reported.

## Efficacy

**Patient traits.**   An overview of the included population's geographic distribution is illustrated in Fig 2. The study population comprised 4046 patients, with a mean age between 19 and 80 years. With respect to the indications, 31 studies focused on periorbital rejuvenation

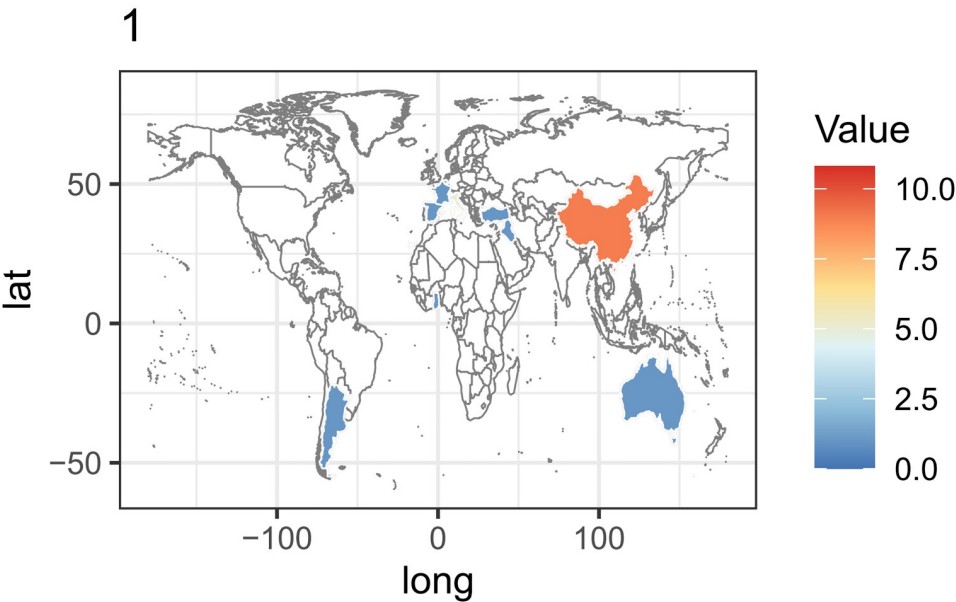

**Fig 2. Geographical distribution of publications and patients.**

while the remaining studies paid close attention to tackling the diseases via reconstructive surgery. Specifically, fat grafting was mainly used to treat patients with aging eyelids, tear trough deformity, and sunken upper eyelids for aesthetic purpose [3–6, 8, 20–30]. Moreover, the primary reconstructive indications included thyroid-associated orbitopathy and anophthalmic sockets [18, 24, 31, 32].

**Patient satisfaction and/or surgeon satisfaction.** The overall satisfaction was relatively comprehensive evaluation toward the efficacy of the fat grafting procedure. Twenty-seven studies paid attention to assessing this outcome and all of them reported a high rate of 90.9% (95% CI, 86.4%–94.0%) (Fig 3). Almost all patients in the above studies stated that they were pleased or very pleased with the cosmetic outcome after the fat grating procedure. Twenty-three studies were assessed by patients with a satisfaction rate of 91.5% (95% CI, 86.7%–94.7%) (Fig 4), while four studies were assessed by patients and surgeons with a result of 81.9% (95% CI, 73.3%–88.2%) (Fig 5). Furthermore, in accordance to the subgroup analysis, the satisfaction rate of specially processed fat graft was lower at 90.3% (95% CI, 79.3%–95.8%) than that of regular fat graft at 91.0% (95% CI, 86.5%–94.1%) (Fig 6). Most of the satisfaction rates were appraised by the preoperative and postoperative photographs, while five studies used a relatively object scale to evaluate the effect and reported a certain score at the end of the treatment. One study conducted by Huang et al. [22] evaluated the cosmetic results using the 5-point Likert scale (1, very unsatisfied; 2, unsatisfied; 3, neutral; 4, satisfied; 5, very satisfied), and the mean score awarded for patients was 4.702, which was significantly higher than the midpoint value of 3 (average) on the 5-point scale, indicating that patients were mostly satisfied with their overall postoperative improvement. Karataş et al. [33] evaluated patients' satisfaction by a questionnaire graded from 1 (not satisfied), 2 (mildly satisfied), 3 (moderately satisfied), and 4 (very satisfied). Grades 1 and 2 were accepted as dissatisfaction and grades 3 and 4 were accepted as satisfaction for the data analysis. Roh et al. [34] employed a grading scale at 3 months, which was completed by an independent medical observer, utilizing an ascending

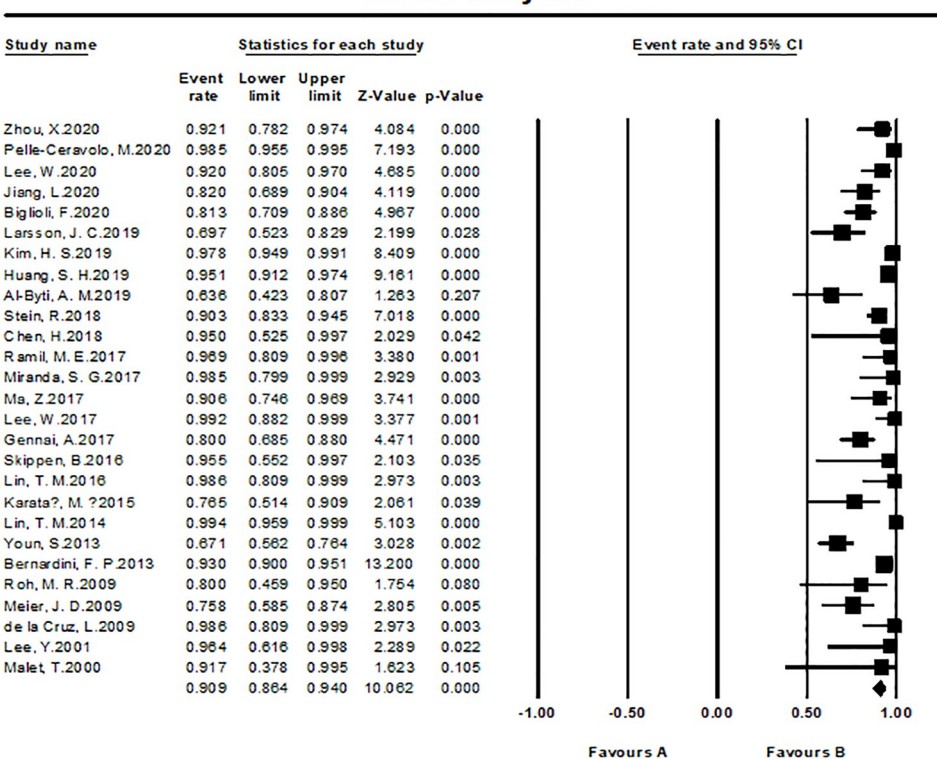

**Fig 3. Meta-analysis-satisfaction rates of all included studies.**

scale ranging from 0 to 4. The average score of their patients was 78%. Youn et al. [35] evaluated their results for dark circle correction by using the Fitzpatrick scale (grades 1–6). During this study, patients were divided into three groups (worse, no change, and improved), and the final graft results were 67.1%, 28%, and 4.9%, respectively. Kim et al. [21] evaluated their results using the modified Goldberg score. The final outcome indicated that major improvements were made in the orbital fat prolapse (preoperative: 1.94 [0.63]; postoperative: 0.07 [0.21]), tear trough depression (preoperative: 1.61 [0.75]; postoperative: 0.33 [0.42]), skin transparency (preoperative: 1.15 [0.97]; postoperative: 0.22 [0.37]), and triangular malar mound (preoperative: 0.37 [0.61]; postoperative: 0.34 [0.58]).

## Safety

**Follow-up.** There existed a broad range of mean follow-up times. During the follow-up period, the patients were evaluated either by imago-logical examination or by preoperative and postoperative photography, or else, a combination of both. In almost all studies, photography was conducted by diverse independent examiners, such as the operating surgeon, an independent plastic surgeon with no working relationship with the primary surgeon, and a secretary from the administrative department of the hospital [36].

**Complications.** As already known, the fat grafting technique is an invasive procedure; inevitably, it will trigger a variety of complications, like any other surgical technique. Many

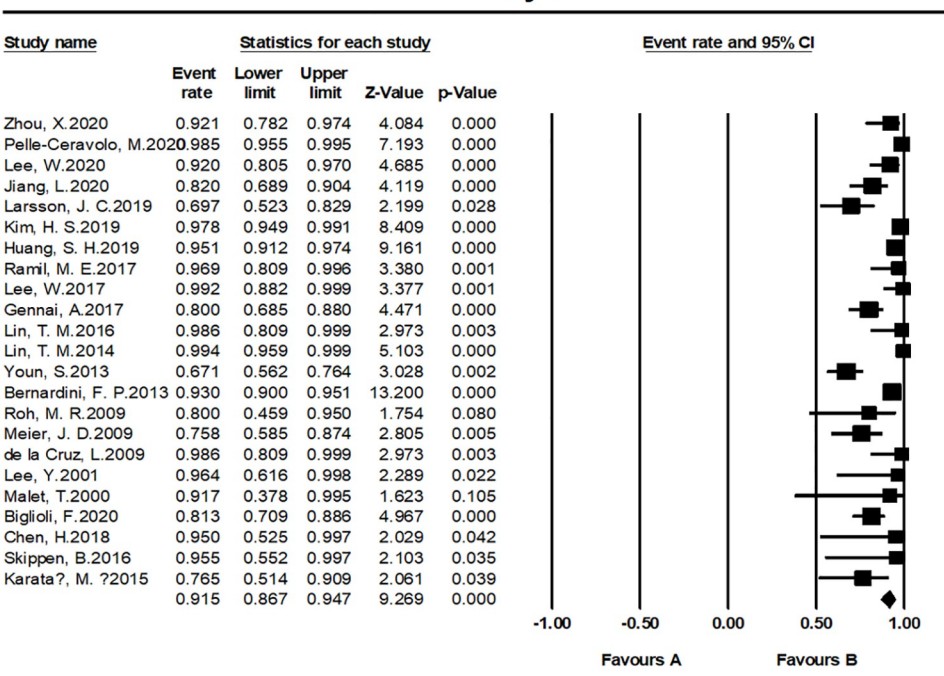

**Fig 4. Meta-analysis-patient satisfaction rates of all included studies.**

complications occurring with fat grafting were minor. As shown in Fig 7, the top five complications were edema, chemosis, contour irregularity, deep wrinkles, and volume excess. The complication rate of reconstructive surgeries was 23.0% (95% CI, 10.6%–42.8%) compared to aesthetic surgeries, which had a complication rate of 6.1% (95% CI, 3.4%–10.6%) (Figs 8 and 9). Furthermore, subgroup analysis showed that the complication rate of specially processed fat graft was lower at 5.1% (95% CI, 2.1%–11.5%) than that of regular fat graft at 10.0% (95% CI, 5.6%–17.4%) (Fig 10). Complications included prolonged swelling, postoperative bruising,

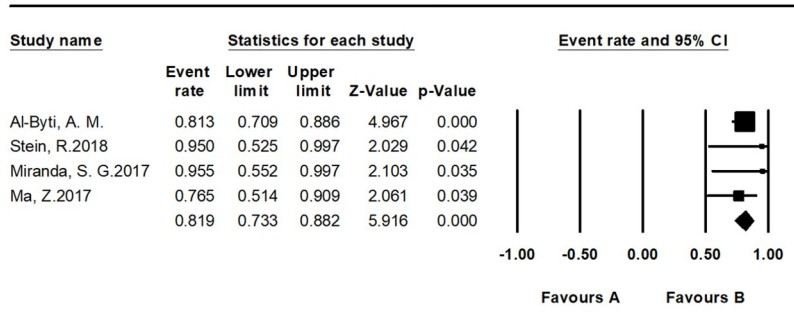

**Fig 5. Meta-analysis-patient and surgeon satisfaction rates of all included studies.**

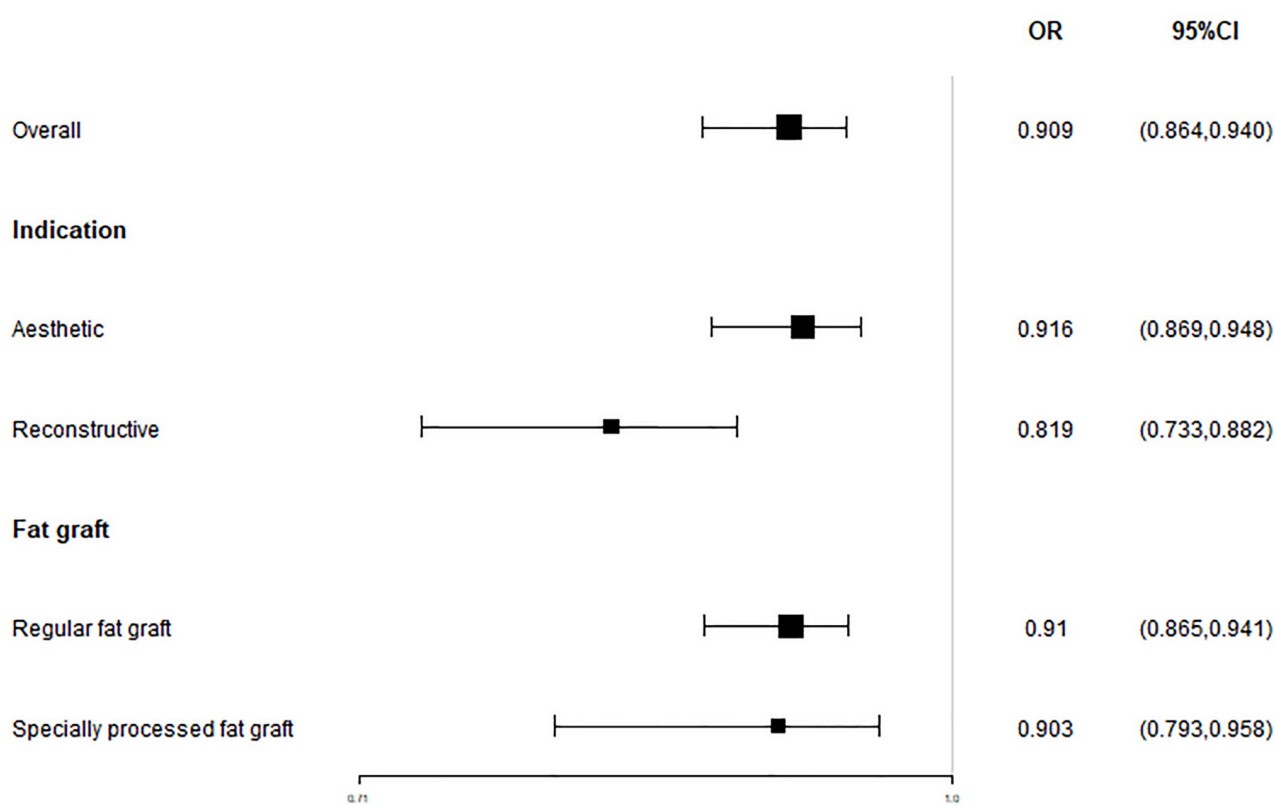

**Fig 6. Subgroup analysis for the pooled estimates of the satisfaction rate (different indications and fat graft treatment method).** The random effects model was applied to minimize heterogeneity.

and edema, which could be alleviated by the patented micro-controlling system of MAFT-- Gun. It could reduce the back-and-forth movements of the injection cannula during the AFG procedure [6]. Periorbital lipogranuloma was identified by Park et al. [37] and surgical excision and intralesional triamcinolone injection were performed to treat this complication; sometimes, just a simple observation can also work owing to the occurrence of spontaneous resolution. More serious complications can occur as well; for example, Essuman et al. [18] showed infection with or without necrosis, which could be treated by antibiotic therapy, specifically, a combination of Guttae Ciprofloxacin 0.3% and Oc. Tetracycline. Ptosis could be treated by grafted fat removal (with or without levator aponeurosis advancement) [38]. After excision of the mass, the symptoms disappeared completely.

## The results of meta-analysis

**Meta-analysis of satisfaction rates.**   We tested the heterogeneity of satisfaction rates, which showed a result of $I^2 = 81.464$ (p < 0.001), suggesting that the research results for the 27 papers were heterogeneous. Thus, a random effects model was used to merge the data for meta-analysis. Meta-analysis of the categorical data revealed an overall proportion of satisfied patients at 91.5% (95% CI, 86.7%–94.7%). With respect to patient satisfaction, a relatively low proportion of plastic surgeons were satisfied with the result at 81.9% (95% CI, 73.3%–88.2%). The satisfaction rate noted with the cosmetic operation was 91.6% (95% CI, 86.9%–94.8%)

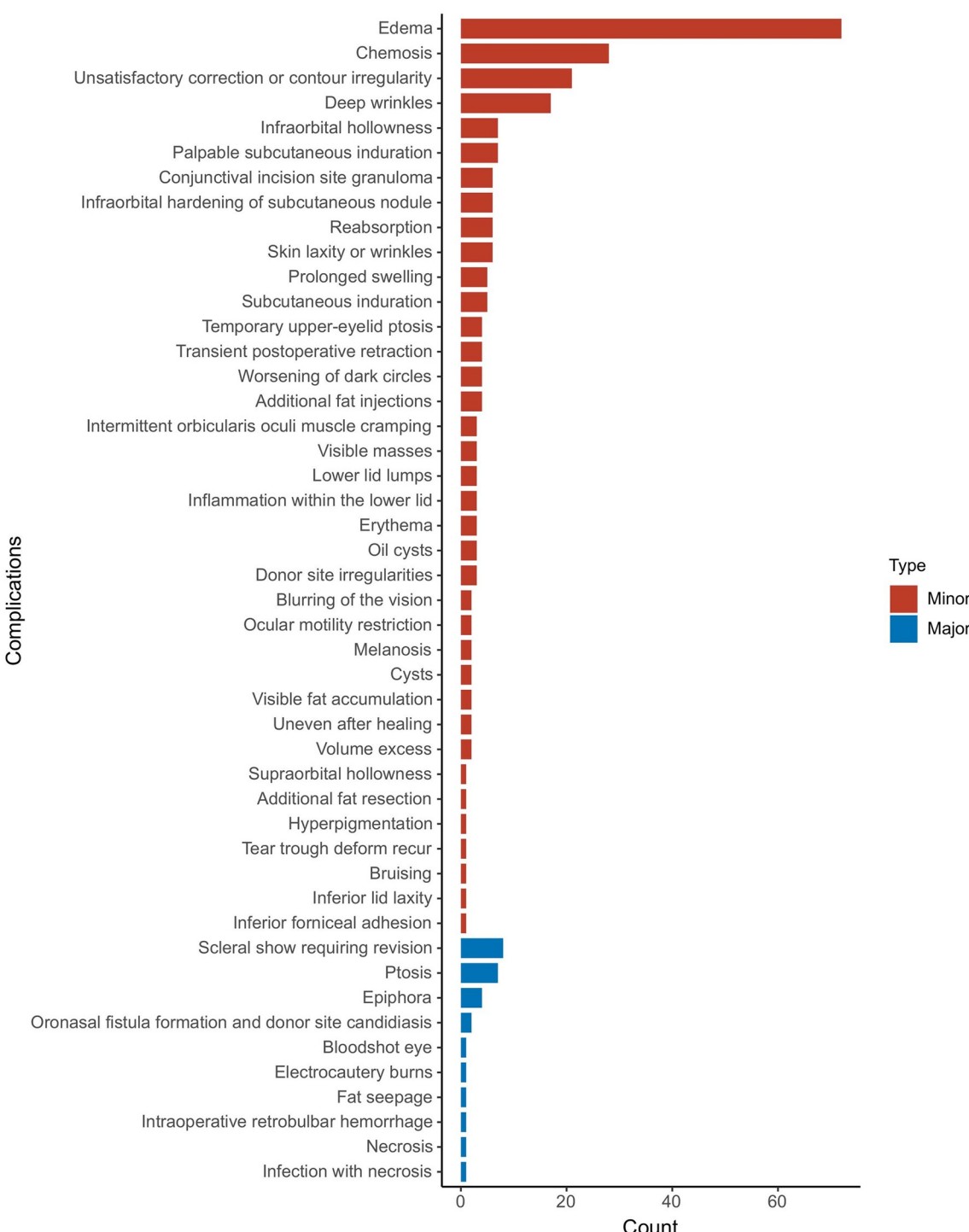

**Fig 7. Complications reported in the literature following periorbital fat augmentation.**

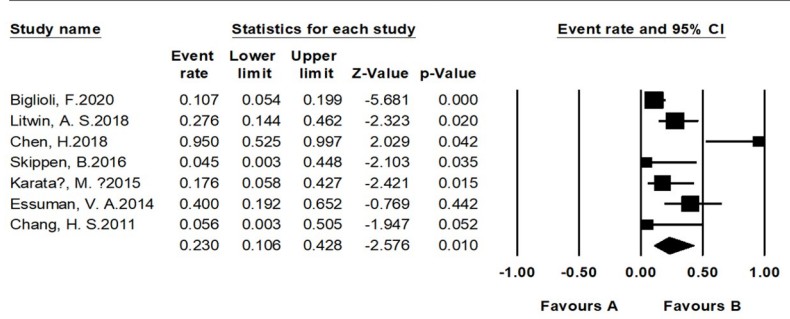

**Fig 8. Meta-analysis-complication rates of all included studies for reconstructive purpose.**

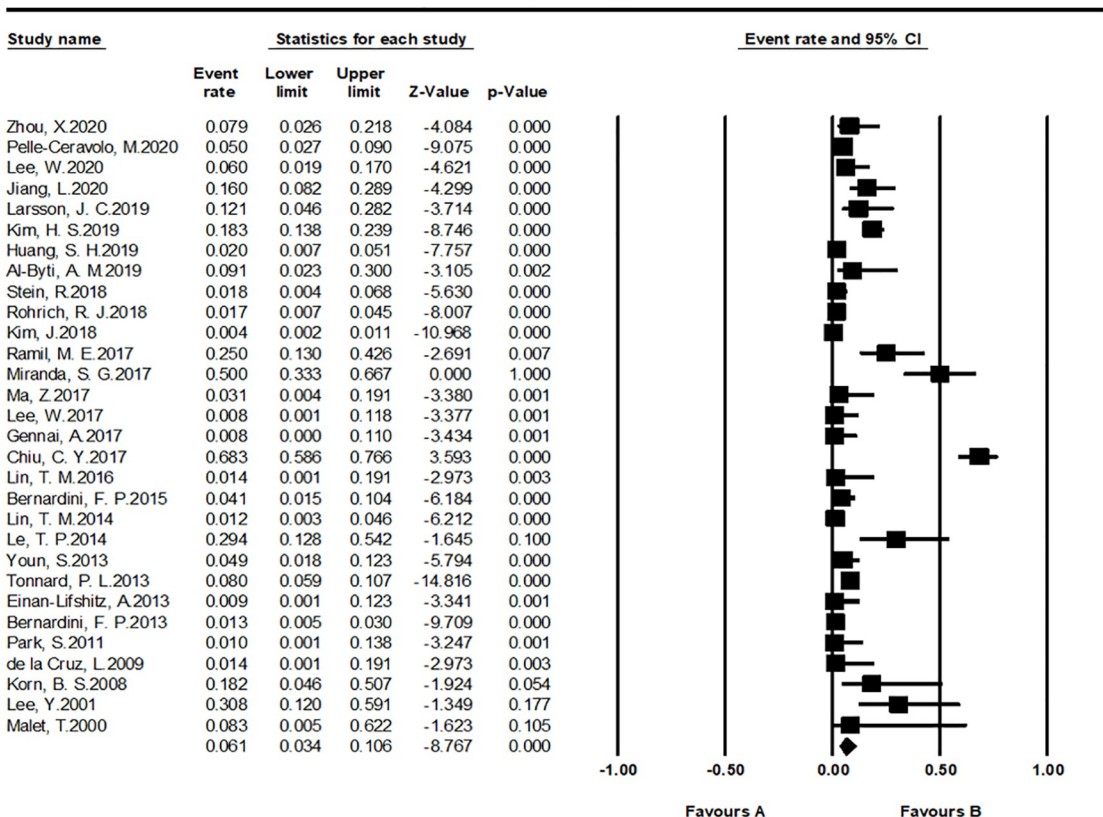

**Meta Analysis**

**Fig 9. Meta-analysis-complication rates of all included studies for aesthetic purpose.**

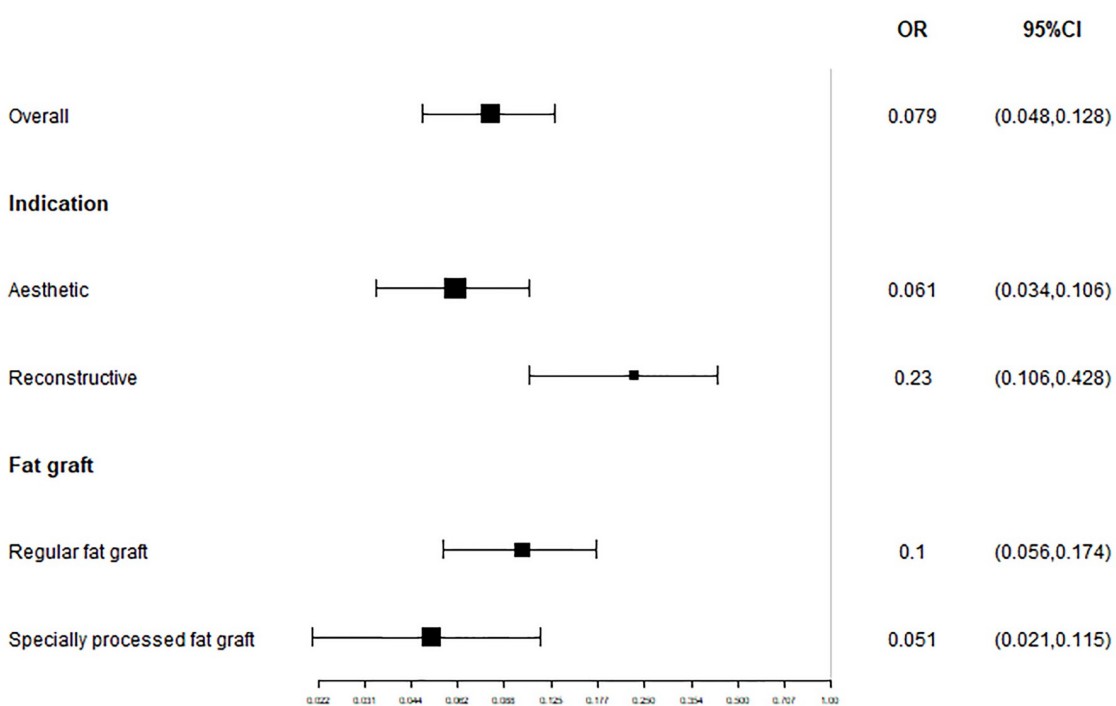

**Fig 10. Subgroup analysis for the pooled estimates of the complication rate (different indications and fat graft treatment method).** The random effects model was applied to minimize heterogeneity.

(Fig 11), while the rate with reconstructive surgeries was 81.9% (95% CI, 73.3%–88.2%) (Fig 12).

**Meta-analysis of complication rates.** We tested the heterogeneity of complication rates, showing a result of I^2 = 91.491 (p <0.001), revealing that the research results for the 37 papers were heterogeneous. Therefore, a random effects model was used to merge the data for meta-analysis. As shown by the forest plot in Fig 13, the complication rate among the included patients aged 19–80 years was 7.9% (95% CI, 4.8%–12.8%). The complication rate reported for cosmetic surgeries was 6.1% (95% CI, 3.4%–10.6%) while the rate evaluated for reconstructive operation was 23.0% (95% CI, 10.6%–42.8%).

## Discussion

Over the years, AFG has gained increased recognition in plastic, reconstructive, and aesthetic eyelid surgeries owing to its prominent advantages compared to conventional treatment, and it has provided an approach with a minimally invasive method and naturally, with less pain and complications [15]. By lipofilling, surgeons can use fat extracted from the hip, abdomen, or inner thigh to reshape and fill up a sunken eyelid. The most extensively used technique was the standardized Coleman. However, not all studies were devoted to this technique; they only made several dedicated modifications. Pelle-Ceravolo Mario and Angelini Matteo diluted 70% fat with saline and infranatant fluid for the purpose of making fat more compatible with the texture of the periocular position [36]. A recent publication by Rohrich et al. showed that the lipoaspirate should be approximately placed in a centrifuge for no longer than 1 minute (2250 rpm) at low pressure to increase the quantity and viability of adipose-derived mesenchymal stem cells to improve the skin quality [39]. Gennai et al. [27] proposed that fat derived from a

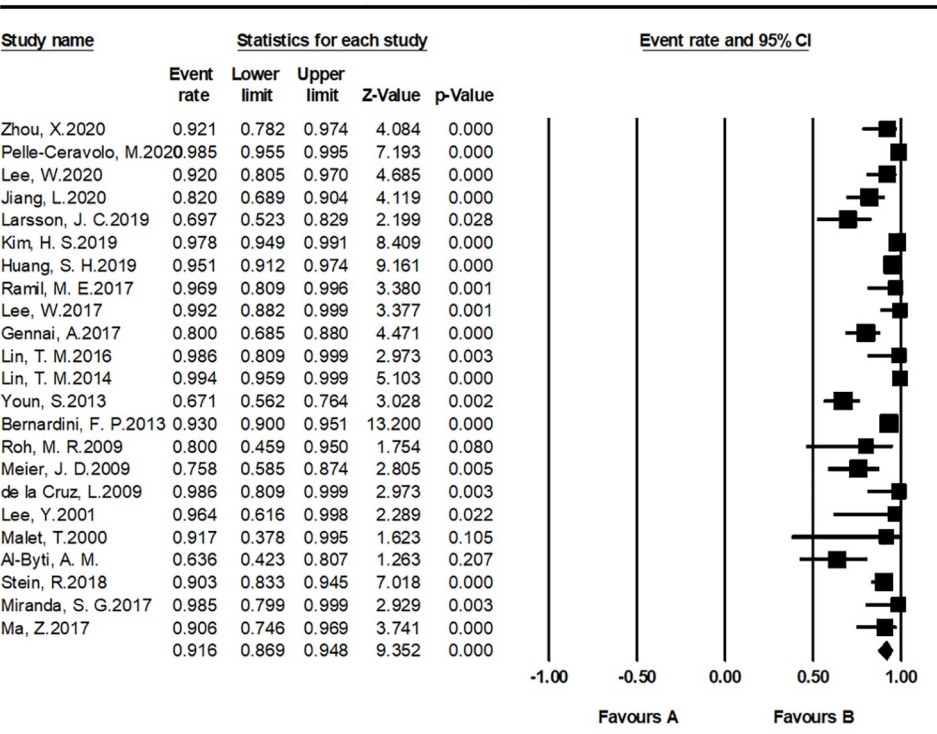

**Fig 11. Meta-analysis-satisfaction rates of all included studies for aesthetic purpose.**

cannula with the smallest port (0.5 mm) showed increased efficacy and viability evaluation of fat harvested with an extremely small side port (0.3 mm) cannula with respect to the correction of aging/thin skin in the periocular region.

The high demand was being dampened mainly by uncertainty concerning anatomical safety due to its special location, which has restricted its application in recent years. Previous studies focusing on this matter did not include RCTs for practical and ethical concerns. One

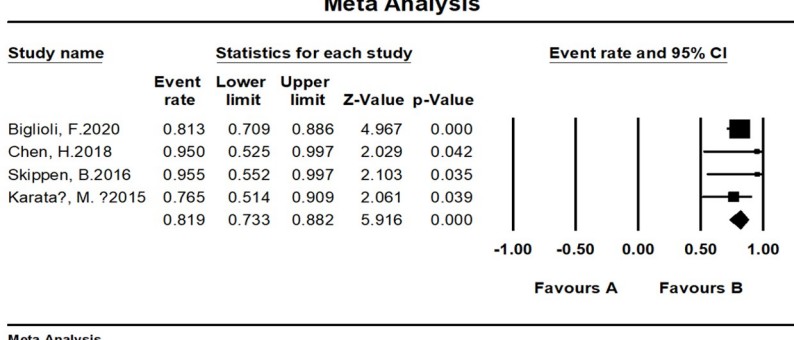

**Fig 12. Meta-analysis-satisfaction rates of all included studies for reconstructive purpose.**

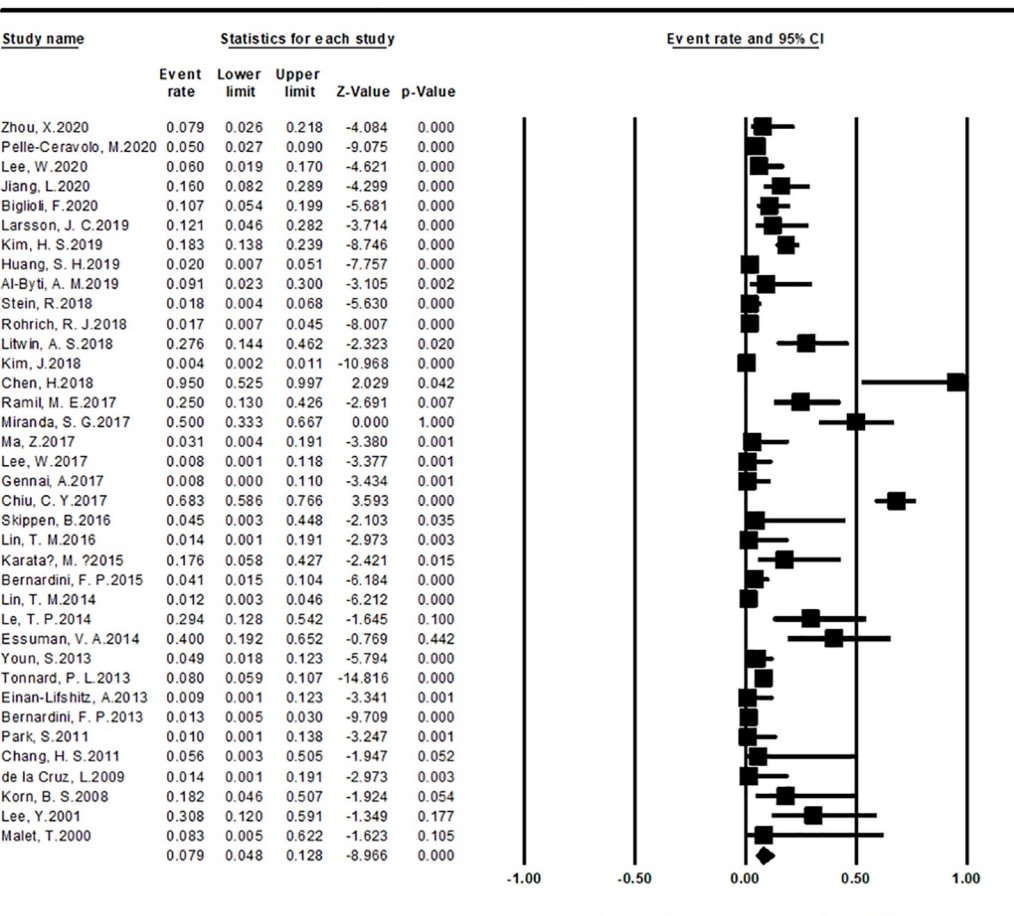

## Meta Analysis

**Fig 13. Meta-analysis-complication rates of all included studies.**

systematic review written by Boureaux et al in 2016 discussed the indications, operative technique, and complications of eyelid fat grafting [12]. It incorporated data from 16 articles that reported AFG utilization in patients with eyelid problems. Almost all of the included articles were case series and only five studies were case reports. Thus, we updated the evidence. To some extent, adipocyte survival is dependent upon nutrients delivered around the periphery of the fat graft, and if the central graft is too far away from the vessels, it will die [40]. The current research focuses on tissue regeneration that includes the use of additive agents, enhancers, or scaffolds to fat. For example, the combination of adipose stem cells (ASCs) and vascular endothelial growth factor (VEGF) can promote neoangiogenesis, and reduce inflammation and local tissue fibrosis [41]. With respect to nanofat injection, it discards the dead adipocyte fraction and injects the purified stromal vascular fraction only to rejuvenate the periocular skin [42]. Moreover, SVF-gel (Stromal Vascular Fraction) has higher ASC and other SVF cell

density than Coleman fat [43]. Thus, it can be a good alternative. More in vitro research needs to be performed as the underlying mechanism behind fat grafting is still not adequately clear.

There were several limitations to this study. First, nearly all of the articles in this meta-analysis were case series, coupled with limited cohort studies. There was a lack of access to RCTs, as setting RCTs might be viewed as unnecessary or unethical. Thus, we conducted a scale to evaluate the included literatures more objectively. Additionally, few studies managed to include a control group because doctors had not invented a suitable and safe alternative to AFG. Second, another limitation was that the follow-up period seemed to present a lot of variability. Almost every study had its own way to decide the length of the follow-up. We concluded that the maximal time for follow-up was 96 months, while the minimal time was 3 months. According to the already published theory, it will take 3 to 6 months for the graft to reach a steady state; therefore, keeping a longer follow-up period is of great importance [44, 45]. Third, notwithstanding the renowned Coleman technique, there is no standard guideline to be followed concerning fat grafting. In particular, it has been a contentious issue for a long period, especially when it comes to the rotational speed, the time of centrifugation, and storage [46]. Also, there is still some debate about the objective standards that could assess the validity of included studies. In this respect, the 5-point Likert scale that was used to appraise patients' satisfaction could only be found in limited studies [22]. Compared to breast fat grafting, a validated outcome scale like Breast-Q is lacking; similarly, there is a lack of effective imago-logical examination, no matter whether it is MRI, CT, or 3D scanner, to measure the actual eyelid volume [47]. For a long time, investigators have been dependent on various subjective questionnaires, preoperative and postoperative photographs, and manmade scales, producing a wide variety of unconvincing results. Specifically, we ought to set an evaluation standard for comprehensive and objective assessment of all the included literature. Finally, we did not track databases in grey literature, which may cause a publication bias and affect the integrity of the data.

## Conclusions

The loss of periorbital volume is an important component of aging, for which AFG is the ideal form of soft tissue replacement. Therefore, it will become a suitable technique for reshaping the eyelids as a standardized method. This meta-analysis reveals that the overall patient satisfaction is relatively high, ranging from 86.4% to 94.0%. With respect to the complications, most of them were minor ones, which can be treated easily or may disappear spontaneously. There was no high rate of severe complications. We further recommend that objective tools assessing the fat retention rate should be invented. Moreover, research hotspots, such as SVF, ACS, and PRP (Platelet Rich Plasma), will open new doors in regenerative and reconstructive surgery.

## Supporting information

**S1 File.**
(ZIP)

## Author Contributions

**Conceptualization:** Fan Yang, Jing Li.

**Data curation:** Zhaohua Ji, Ting Fu.

**Formal analysis:** Liwei Peng.

**Methodology:** Kun Liu.

**Software:** Fan Yang, Wenjie Dou.

**Supervision:** Yuejun Li, Yong Long, Weilu Zhang.

**Validation:** Liwei Peng.

**Writing – original draft:** Fan Yang.

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
