## [Decision Letter · Decision Letter 0]

7 Oct 2020

PONE-D-20-28837

Efficacy, safety and complications of autologous fat grafting for rejuvenation of eyelids and periorbital area:  A systematic review and meta-analysis

PLOS ONE

Dear Dr. Zhang,

Thank you for submitting your manuscript to PLOS ONE. After careful consideration, we feel that it has merit but does not fully meet PLOS ONE’s publication criteria as it currently stands. Therefore, we invite you to submit a revised version of the manuscript that addresses the points raised during the review process.

We look forward to receiving your revised manuscript.

Kind regards,

Fabio Santanelli, di Pompeo d'Illasi, MD, PhD

Academic Editor

PLOS ONE

Journal Requirements:

3. At this time, we ask that you please provide the full search strategy and search terms for at least one database used as Supplementary Information.

4. Please provide the date ranges for the database searches you performed.

5. Please attach a Supplemental file of the results of the individual components of the quality assessment, not just the overall score, for each study included. Please also explain the reasons, and number of studies excluded for each reason, in the flow diagram. Thank you.

"This study was funded by National Natural Science Foundations of China (81773488, 81772096 and 81803289)"

"No"

7. In your Data Availability statement, you have not specified where the minimal data set underlying the results described in your manuscript can be found. PLOS defines a study's minimal data set as the underlying data used to reach the conclusions drawn in the manuscript and any additional data required to replicate the reported study findings in their entirety. All PLOS journals require that the minimal data set be made fully available. For more information about our data policy, please see http://journals.plos.org/plosone/s/data-availability.

8. We note that Figure 2 in your submission contain map images which may be copyrighted. All PLOS content is published under the Creative Commons Attribution License (CC BY 4.0), which means that the manuscript, images, and Supporting Information files will be freely available online, and any third party is permitted to access, download, copy, distribute, and use these materials in any way, even commercially, with proper attribution. For these reasons, we cannot publish previously copyrighted maps or satellite images created using proprietary data, such as Google software (Google Maps, Street View, and Earth). For more information, see our copyright guidelines: http://journals.plos.org/plosone/s/licenses-and-copyright.

8.1.    You may seek permission from the original copyright holder of Figure 2 to publish the content specifically under the CC BY 4.0 license. 

8.2.    If you are unable to obtain permission from the original copyright holder to publish these figures under the CC BY 4.0 license or if the copyright holder’s requirements are incompatible with the CC BY 4.0 license, please either i) remove the figure or ii) supply a replacement figure that complies with the CC BY 4.0 license. Please check copyright information on all replacement figures and update the figure caption with source information. If applicable, please specify in the figure caption text when a figure is similar but not identical to the original image and is therefore for illustrative purposes only.

Reviewers' comments:

Reviewer's Responses to Questions

**Comments to the Author**

1. Is the manuscript technically sound, and do the data support the conclusions?

Reviewer #1: Yes

Reviewer #2: Yes

2. Has the statistical analysis been performed appropriately and rigorously? 

Reviewer #1: Yes

Reviewer #2: Yes

3. Have the authors made all data underlying the findings in their manuscript fully available?

Reviewer #1: Yes

Reviewer #2: Yes

4. Is the manuscript presented in an intelligible fashion and written in standard English?

Reviewer #1: Yes

Reviewer #2: Yes

5. Review Comments to the Author

Reviewer #1: I have read with great interest the manuscript entitled "Efficacy, safety and complications of autologous fat grafting for rejuvenation of eyelids and periorbital area: A systematic review and meta-analysis". These are my comments to the authors.

1) In the abstract although the authors introduce the abbreviation AFG for autologous fat grafting, they go back to using autologous fat grafting and not AFG in their conclusion. Please check the manuscript and correct this minor error

2) In the abstract in the conclusion section they report that "AFG might be....and reconstruction) the title of the manuscript refers to evaluation of AFG in rejuvenation, i think the connection of AFG to the reconstruction of the periorbital region is not appropriate. I advice to only use the periorbital rejuvenation and contour defects.

3)Line 56 the word isolation is misspelled as insolation

4)In the exclusion criteria line 85-86 the authors refer to patients with a history of other eyelid surgery but then give an example of periorbital rejuvenation and "reconstruction" with hyaluronic acid. Hyaluronic acid is used for rejuvenation and contour correction not for reconstruction and it is not a surgical procedure. The authors should specify and make this exclusion criteria clear to the read as to what they mean.

5) Line 473 a citation is mentioned as INVALID CITATION. The authors should check the references for errors

6) The article is generally well written but minor reviewing of language is needed.

Reviewer #2: thank you very much to give me the opportunity to review this article. the authors made a systematic review using international guidelines and it is correct. i think that the meta anaysis is poor and did not support the conclusions. it is important to do a meta analysis with high methods to support the high quality of the journal and the topic treated.

i suggest to made a flow chart diagram of the review.

6. PLOS authors have the option to publish the peer review history of their article (what does this mean?). If published, this will include your full peer review and any attached files.

Reviewer #1: No

Reviewer #2: No

---

## [Author Response · Author response to Decision Letter 0]

30 Nov 2020

Dear Editor,

We would like to express sincere gratitude for your letter and advice. We have revised the manuscript in accordance with the point raised by the academic editor and reviewers. Point-to-point answers have been provided below. 

Manuscript title: Efficacy, safety and complications of autologous fat grafting for rejuvenation of eyelids and periorbital area: A systematic review and meta-analysis

Academic editor

Answer: Thank you for your comment. We have checked manuscript to meet the style requirements. 

Answer: Thank you very much for your advice. We have sought help from the professional scientific editing service. Here is our Certi¬ficate of English Language Editing.

3. At this time, we ask that you please provide the full search strategy and search terms for at least one database used as Supplementary Information.

Answer: Thank you very much for your advice. We have revised in the Supporting Information files. Specifically speaking, we provide the full search strategy and search terms for PubMed and Cochrane library in details. The following highlight parts have been added as the supplementary information.

PubMed

#1 ((((((((((fat grafting) OR (lipograft)) OR (lipoinjection)) OR (lipotransfer)) OR (fat transfer)) OR (fat transplant)) OR (lipostructure)) OR (lipofilling)) OR (fat injection)) OR (lipomodeling)) OR (fat transplantation) Sort by: Most Recent 23199

#2 (eyelid) OR (periocular) Sort by: Most Recent 49264

#3 #1 AND #2 324

Cochrane 

(fat grafting OR lipograft OR lipoinjection OR lipotransfer OR fat transfer OR fat transplant OR lipostructure OR lipofilling OR fat injection OR lipomodeling OR fat transplantation) in All Text AND (eyelid OR periocular) in All Text - (Word variations have been searched) 10

The date ranges for the database searches (before November 1, 2019)

4. Please provide the date ranges for the database searches you performed.

Answer: Thank you very much for your advice. Likewise, we have revised in the Supporting Information files. We provide the data ranges for the database searches. The following highlight parts have been added as the supplementary information. 

PubMed

#1 ((((((((((fat grafting) OR (lipograft)) OR (lipoinjection)) OR (lipotransfer)) OR (fat transfer)) OR (fat transplant)) OR (lipostructure)) OR (lipofilling)) OR (fat injection)) OR (lipomodeling)) OR (fat transplantation) Sort by: Most Recent 23199

#2 (eyelid) OR (periocular) Sort by: Most Recent 49264

#3 #1 AND #2 324

Cochrane 

(fat grafting OR lipograft OR lipoinjection OR lipotransfer OR fat transfer OR fat transplant OR lipostructure OR lipofilling OR fat injection OR lipomodeling OR fat transplantation) in All Text AND (eyelid OR periocular) in All Text - (Word variations have been searched) 10

The date ranges for the database searches (before November 1, 2019)

5. Please attach a Supplemental file of the results of the individual components of the quality assessment, not just the overall score, for each study included. Please also explain the reasons, and number of studies excluded for each reason, in the flow diagram. Thank you.

Answer: Thank you very much for your advice and we agree with you. We have given the individual scores for each study included, which will provide studies a clear quality assessment. The revised edition can be found in the Supporting Information files. And the flow diagram has been done in the Figure files.

NR: not reported

"This study was funded by National Natural Science Foundations of China (81773488, 81772096 and 81803289) We note that you have provided funding information that is not currently declared in your Funding Statement. However, funding information should not appear in the Acknowledgments section or other areas of your manuscript. We will only publish funding information present in the Funding Statement section of the online submission form.

Answer: Thank you very much for your advice. We apologize for the confusion regarding the funding statement. We have revised our Funding Statement in the online submission form.

7. In your Data Availability statement, you have not specified where the minimal data set underlying the results described in your manuscript can be found. PLOS defines a study's minimal data set as the underlying data used to reach the conclusions drawn in the manuscript and any additional data required to replicate the reported study findings in their entirety. All PLOS journals require that the minimal data set be made fully available. For more information about our data policy, please see http://journals.plos.org/plosone/s/data-availability.

Answer: Thank you very much for your advice. We have provided the minimal data set in the Supporting Information files. We are very supportive of the Data Availability statement. So, we provide the raw data as clearly as possible with the belief that this behavior will promote academic development. 

8. We note that Figure 2 in your submission contain map images which may be copyrighted. All PLOS content is published under the Creative Commons Attribution License (CC BY 4.0), which means that the manuscript, images, and Supporting Information files will be freely available online, and any third party is permitted to access, download, copy, distribute, and use these materials in any way, even commercially, with proper attribution. For these reasons, we cannot publish previously copyrighted maps or satellite images created using proprietary data, such as Google software (Google Maps, Street View, and Earth). For more information, see our copyright guidelines: http://journals.plos.org/plosone/s/licenses-and-copyright.

Answer: Thank you very much for your advice. We apologize for the confusion made in the Figure 2. Without taking the copyright into full consideration, we use the map downloaded from the internet. We have made a new “geographic distribution” map by the software “Hiplot”. The new picture can be found in the Figure files.

Reviewer #1: I have read with great interest the manuscript entitled "Efficacy, safety and complications of autologous fat grafting for rejuvenation of eyelids and periorbital area: A systematic review and meta-analysis". These are my comments to the authors.

1) In the abstract although the authors introduce the abbreviation AFG for autologous fat grafting, they go back to using autologous fat grafting and not AFG in their conclusion. Please check the manuscript and correct this minor error

Answer: Thank you very much for your advice. We apologize for the mistake made in the manuscript. We have revised the error, undoubtedly, this is our mistake, thanks for reminding us. 

2) In the abstract in the conclusion section they report that "AFG might be....and reconstruction) the title of the manuscript refers to evaluation of AFG in rejuvenation, i think the connection of AFG to the reconstruction of the periorbital region is not appropriate. I advice to only use the periorbital rejuvenation and contour defects.

Answer: Thank you very much for your advice and we totally agree with you. We apologize for the confusion made in the title. We have revised the title to “Efficacy, safety and complications of autologous fat grafting to the eyelids and periorbital area: A systematic review and meta-analysis”. Indications in the meta-analysis concern a lot of diseases such as aging eyelids, tear trough deformity and anophthalmic reconstruction. The former title cannot give an all-around description of the content. 

3)Line 56 the word isolation is misspelled as insolation 

Answer: Thank you very much for your advice. We apologize for the spelling mistake in the manuscript. We have revised the content. We shouldn’t have made such a simple grammatical mistake.

4)In the exclusion criteria line 85-86 the authors refer to patients with a history of other eyelid surgery but then give an example of periorbital rejuvenation and "reconstruction" with hyaluronic acid. Hyaluronic acid is used for rejuvenation and contour correction not for reconstruction and it is not a surgical procedure. The authors should specify and make this exclusion criteria clear to the read as to what they mean.

Answer: Thank you very much for your advice. And we apologize for the confusion made in the manuscript. In the paper titled “the Tear trough deformity: different types of anatomy and treatment options”, the author incorporated seventy-eight patients with tear trough deformity. Moreover, he rated them as three class. Ten cases in their series were classified as class I, eighteen cases as class II and fifty cases as class III. Patients of class I or class II were treated using hyaluronic acid gel (18 cases) or autologous fat injections (10 cases). The author explicitly differentiated between patients received hyaluronic acid and those who got the autologous fat injection. And we excluded ones treated by hyaluronic acid. So, finally, we contained ten patients who received autologous fat. As you said, hyaluronic acid is neither a surgical procedure nor for reconstruction. I apologize for the mistake made in the inclusion and exclusion criteria. We have changed the inappropriate expression in the manuscript. 

5) Line 473 a citation is mentioned as INVALID CITATION. The authors should check the references for errors

Answer: Thank you very much for your advice. We apologize for the mistake. We have revised in the reference part.

6) The article is generally well written but minor reviewing of language is needed.

Answer: Thank you very much for your advice. Owing to the limited writing skill, we cannot write paper like native speakers. So, we seek help from a professional scientific editing service. We have provided the Certi¬ficate of English Language Editing in the Supporting Information files.

Reviewer #2: thank you very much to give me the opportunity to review this article. the authors made a systematic review using international guidelines and it is correct. I think that the meta analysis is poor and did not support the conclusions. it is important to do a meta analysis with high methods to support the high quality of the journal and the topic treated.

i suggest to made a flow chart diagram of the review.

Answer: Thank you very much for your advice and we have considered your comments carefully. Indeed, there were several limitations in this study. First, nearly all of the articles in this meta-analysis were case series, obviously, low quality. There was a lack of access to RCTs, on account of setting RCTs might be viewed as unnecessary or unethical. So, we conducted quality evaluation in accordance with the international scale, for example, the Ottawa-Newcastle Scale, to evaluate the included literatures more objectively. Besides, we used a blinded method to ensure quality and a third reviewer resolved any disagreements. Second, the follow-up period seemed to present a lot of variability. According to already published theory, it will take 3 to 6 months for graft to reach a steady state. Thus, the studies included are followed over 3 months. Third, there was still debate about objective standards which could assess the validity of included studies. Studies included use a lot of objective scales, such as, 5-point Likert scale, grading scale, Fitzpatrick scale and so on. And we have made a flow chart diagram of the review in the “figure” file.

We thank you for your time and consideration and hope that the modified manuscript is now suitable for publication.

Sincerely yours,

Fan Yang

Department of Plastic Surgery and Burns, 

Fourth Military Medical University, 

Xi’an 710032, Shaanxi, China

Email: crystalyangfan@163.com

---

## [Decision Letter · Decision Letter 1]

23 Feb 2021

PONE-D-20-28837R1

Efficacy, safety and complications of autologous fat grafting to the eyelids and periorbital area: A systematic review and meta-analysis

PLOS ONE

Dear Dr. Zhang,

Thank you for submitting your manuscript to PLOS ONE. After careful consideration, we feel that it has merit but one single point must be addressed before the formal acceptance. Please provide the date ranges for the database searches you performed in the main file of the manuscript (Methods section - Search strategy) and states if any language restriction was applied in the search process. Therefore, we invite you to submit a revised version of the manuscript that addresses the point raised during the review process.

We look forward to receiving your revised manuscript.

Kind regards,

Endi Lanza Galvão

Academic Editor

PLOS ONE

Journal Requirements:

Reviewers' comments:

Reviewer's Responses to Questions

**Comments to the Author**

1. If the authors have adequately addressed your comments raised in a previous round of review and you feel that this manuscript is now acceptable for publication, you may indicate that here to bypass the “Comments to the Author” section, enter your conflict of interest statement in the “Confidential to Editor” section, and submit your "Accept" recommendation.

Reviewer #2: All comments have been addressed

2. Is the manuscript technically sound, and do the data support the conclusions?

Reviewer #2: Yes

3. Has the statistical analysis been performed appropriately and rigorously? 

Reviewer #2: Yes

4. Have the authors made all data underlying the findings in their manuscript fully available?

Reviewer #2: Yes

5. Is the manuscript presented in an intelligible fashion and written in standard English?

Reviewer #2: Yes

6. Review Comments to the Author

Reviewer #2: the authors made the revisions requested and the manuscript improved. i think that the manuscript could be considered for publication in this form.

7. PLOS authors have the option to publish the peer review history of their article (what does this mean?). If published, this will include your full peer review and any attached files.

Reviewer #2: No

---

## [Author Response · Author response to Decision Letter 1]

26 Feb 2021

Dear Editors and Reviewers:

Thank you for your letter and for the reviewers’ comments concerning our manuscript entitled “Efficacy, safety and complications of autologous fat grafting to the eyelids and periorbital area: A systematic review and meta-analysis” (ID: PONE-D-20-28837R1). Those comments are all valuable and very helpful for revising and improving our paper, as well as the important guiding significance to our research. We have studied comments carefully and have made corrections which we hope to be met with approval. And we have uploaded the protocol to the protocols.io according to your requirements. Revised portion are marked with track changes in the paper. The main corrections in the paper are as follows:

Editor: Please provide the date ranges for the database searches you performed in the main file of the manuscript (Methods section - Search strategy) and states if any language restriction was applied in the search process.

Response: We are very sorry for our negligence of data range and language restriction. The specific information has been added to the methods section of the article.

A systematic database search was carried out before November 11, 2020. There were no restrictions with respect to language.

Thank you and best regards.

Yours sincerely,

Weilu Zhang

Email: zhangweilu@126.com

---

## [Editor Report · Decision Letter 2]

1 Mar 2021

Efficacy, safety and complications of autologous fat grafting to the eyelids and periorbital area: A systematic review and meta-analysis

PONE-D-20-28837R2

Dear Dr. Zhang,

We’re pleased to inform you that your manuscript has been judged scientifically suitable for publication and will be formally accepted for publication once it meets all outstanding technical requirements.

Kind regards,

Endi Lanza Galvão

Academic Editor

PLOS ONE
---

## [Editor Report · Acceptance letter]

15 Mar 2021

PONE-D-20-28837R2 

Efficacy, safety and complications of autologous fat grafting to the eyelids and periorbital area: A systematic review and meta-analysis 

Dear Dr. Zhang:

I'm pleased to inform you that your manuscript has been deemed suitable for publication in PLOS ONE. Congratulations! Your manuscript is now with our production department. 

Kind regards, 

on behalf of

Dr. Endi Lanza Galvão 

Academic Editor

PLOS ONE